# Recent Progress in the Theranostics Application of Nanomedicine in Lung Cancer

**DOI:** 10.3390/cancers11050597

**Published:** 2019-04-29

**Authors:** Anubhab Mukherjee, Manash Paul, Sudip Mukherjee

**Affiliations:** 1Department of Translational Neurosciences and Neurotherapeutics, John Wayne Cancer Institute, Providence Saint John’s Health Center, 2200 Santa Monica Boulevard, Santa Monica, CA 90404, USA; 2Division of Pulmonary and Critical Care Medicine, David Geffen School of Medicine, The University of California, Los Angeles (UCLA) Factor Bldg. 10-240, 621 Charles E. Young Dr., Los Angeles, CA 90095, USA; 3Department of Bioengineering, Rice University, 6500 Main Street, Houston, TX 77005, USA

**Keywords:** lung cancer, nanomedicine, theranostics, clinical status, cancer therapy

## Abstract

Lung cancer is one of the leading causes of cancer-related death worldwide. Non-small cell lung cancer (NSCLC) causes around 80% to 90% of deaths. The lack of an early diagnosis and inefficiency in conventional therapies causes poor prognosis and overall survival of lung cancer patients. Recent progress in nanomedicine has encouraged the development of an alternative theranostics strategy using nanotechnology. The interesting physico-chemical properties in the nanoscale have generated immense advantages for nanoparticulate systems for the early detection and active delivery of drugs for a better theranostics strategy for lung cancer. This present review provides a detailed overview of the recent progress in the theranostics application of nanoparticles including liposomes, polymeric, metal and bio-nanoparticles. Further, we summarize the advantages and disadvantages of each approach considering the improvement for the lung cancer theranostics.

## 1. Introduction

Lung cancer is one of the prevalent malignancies and leading causes of cancer-related mortality and the most common cancer in men, accounting for an estimated 154,050 deaths in 2018 and worldwide [1,2,3]. As per the GLOBOCAN, a project of the International Agency for Research on Cancer (IARC), estimation, approximately 18.1 million new cancer cases would have been detected in 2018 [2]. This dreadful disease owes its origin mostly (85%) to long term tobacco smoking [4]. It turns out that at the time of diagnosis lung cancer in many patients has metastasized to other tissues in the body. According to the 2012 global lung cancer statistics, an estimated 1.8 million new cases were reported 58% of which occurred in less developed countries [5]. The American Cancer Society reported over 221,200 estimated new cases of lung cancer and 158,040 death cases in 2015 in the United States [2,3]. Around 85% of lung cancer patients had non-small cell lung cancer while the remaining 15% had small cell lung cancer. The survival rate in lung cancer patients depends primarily on early diagnosis and surgical resection of the tumor tissue is often the preferred therapy [6,7]. Clinically used therapeutic modalities are still associated with a poor outcome: Only <20% five-year overall survival is reported as cancer cells routinely become impervious to drugs. Among all the presently available cancer therapies, chemotherapy is the most widely used treatment strategy for lung cancer. The insufficient drug concentration in the tumor tissue is one of the major impediments retarding the clinical success of lung cancer chemotherapy. To meet this up-hill therapeutic challenge, repeated applications of anti-cancer agents at high concentrations are being used in systemic chemotherapy which is causing adverse side effects. The commonly observed adverse side effects in chemotherapy based cancer treatments mostly originate from the ability of many potent cytotoxic drugs to penetrate non-cancerous healthy body tissues in addition to tumor tissues [8]. Novel therapeutic modalities for more effective treatment of lung cancer are, therefore, urgently needed.

In recent years, enormous endeavors have been directed toward the development of new warheads and their effective carriers to reduce the probability of occurrences of multi-drug resistance, combinations of hydrophilic with hydrophobic drugs, small interfering RNA (siRNAs) with hydrophobic drugs being majorly explored among them [9]. Global efforts are also being witnessed on delivering potent anti-cancer drugs selectively to tumor tissues by encapsulating the drugs within various types of drug carriers [10,11,12]. The exo-surfaces of such drug carriers are covalently grafted with tumor-specific targeting motifs including small peptides, aptamers, proteins, and antibodies [13,14,15,16,17].

Much to our intrigue, theranostics (therapy plus diagnostic) nanomedicine has emerged as a propitious paradigm in cancer therapy. It involves advantages of the both world: highly efficacious nanocarriers to ferry cargo while loading onto them both imaging and therapeutic agents. The idea spawned a plethora of nanoparticles, well suitable for drug delivery as well as diagnosis, facilitating the advent of personalized medicine. There exists four crucial aspects to take into account while designing an efficacious nanoplatform based therapeutics: (i) Selecting a potent therapeutic, ranging from small molecule drugs to larger peptide or nucleic acid; (ii) to choose a stable carrier; (iii) to adopt a targeting and drug release strategy; and (iv) to carefully single out an imaging agent [18,19,20,21,22,23,24].

The present review is an attempt to update major advancements in lung cancer theranostics: A description of development of various nanosystems (liposomes, polymeric nanoparticles, metal nanoparticles, bio-nanoparticles, etc.) for efficient delivery of an array of theranostics in lung cancer.

## 2. Lung Cancer: Category, Cause, Molecular Target, and Limitations of Conventional Therapy

Non-small cell lung cancer is the most prevalent cancer and can be further categorized into two major subtypes which include small cell lung cancer (SCLC) and non-small cell lung cancer (NSCLC) based on the histological appearance. Though SCLC is more aggressive but less frequent as compared to NSCLC. NSCLC can be sub-classified into three major histological subtypes: Adenocarcinoma, squamous cell-carcinoma, and large cell-lung cancer. Again, each of the subtypes is distinct and responds differently to available therapies.

Lung carcinogenesis is influenced by the interaction of ecological factors, for example, tobacco smoke, and genetic susceptibility. The cancer susceptibility increases significantly with rare germ line mutations like p53, retinoblastoma, epidermal growth factor receptor (EGFR), etc. Moreover, reduced DNA repair efficiency may also play a critical role in lung carcinogenesis [25]. Chemicals in tobacco smoke are reported to play a major role in lung carcinogenesis. Research advances have provided clear evidence for the role of tyrosine kinases in the pathophysiology of lung cancer. Constitutive kinase activation and downstream signaling may arise due to mutation(s), overexpression, and autocrine paracrine incitement, leading to cancer. Oncogenic activation of tyrosine kinase like EGFR, PIK3CA, MET etc. are frequently observed in NSCLC and thereby offer opportunity for therapeutic targeting [25,26].

Despite the fact that progresses are made in the treatment of non-small cell lung cancer (NSCLC), the five-year survival rate for lung malignant growth has expanded by just 5% in recent years. Surgery serves an important treatment modality of early stage disease but surgery for lung cancer is complex and can have serious consequences. Early detection of lung malignant growth is another hurdle in effective treatment of lung cancer patients. Roughly seventy five percent of patients with lung disease will present symptoms, and the majority of these have an advanced stage of tumor at the time of diagnosis.

The emergence of cancer chemotherapy and radiation therapy is partially effective in the initial stage of NSCLC treatment. Due to the advent of robust sequencing techniques and initiation of large genome wide association studies it is quite clear that molecular heterogeneity exists even in the same cancer subtypes. Heterogeneity can exist among the primary tumor and the metastatic counterpart, even within the cells of a particular tumor or based on the cell of origin. The molecular diversity and the cancer cells ability to acquire adaptive resistance create a definite challenge in planning an effective therapy [27]. A new approach addressing patient specific molecular specificities and patient centered approach is needed.

## 3. Alternative Theranostics by Nanomedicine

Nanotechnology is one of the fast growing fields in the area of biomedical science that has been utilized to solve different biological problems including therapeutics and diagnostics. Recently, nanotechnology has been widely utilized for the treatment of various diseases including cancer, diabetes, bacterial infections, cardiovascular diseases, etc. [28,29,30]. Due to several limitations in the conventional therapeutic strategies for lung cancers, scientists and researchers have focused on the development of the nanoscale therapeutic agents, whereas the delivery system includes liposomal nanoparticles, polymeric nanoparticles, metal nanoparticles, and bio-nano particles. The lung cancer theranostics applications of these nanoparticles has been largely effective due to their small size, that enables them to specifically accumulate in tumor cells due to an enhanced permeability and retention effect (EPR) [31]. Moreover, nanoparticles are easy to functionalize and demonstrates high drug loading due to large surface area to volume [29]. Apart from that, due to good biocompatibility and the capability of overcoming clearance by the kidney effacing long circulation nanoparticles holds edge over conventional therapeutic treatments. Moreover, several nanoparticles displayed multifunctional abilities like imaging, diagnostics, therapeutics, sensing that helps researchers to utilize these nanomaterials for multifunctional biomedical applications in lung cancer theranostics. Here in this review, we have focused on the theranostics applications of various types of nanomaterials that have shown enormous promise for the applications in lung cancer diagnosis and therapy including (a) liposomal nanoparticles, (b) polymeric nanoparticles, (c) bio-nanoparticles, and (d) metal nanoparticles (Figure 1).

## 4. Liposomal Nanoparticles for Lung Cancer Theranostics

Liposomes are artificially generated vesicles with a bilayer structure spontaneously formed when natural or synthetic amphiphatic lipids get dispersed in water. Ever since their inception, they have largely been explored as drug delivery vehicles because of their biocompatibility and beneficial safety profile. The bilayer structure of liposomes comprises of phosphatidylcholine, cholesterol, etc. and they are known to carry an array of small molecule and large molecule therapeutics of hydrophobic and hydrophilic origin. Their surface can be modified by grafting polyethylene glycol (PEG), which prolongs their half-life in circulation [32,33]. Doxil and Myocet, two leading Doxorubicin liposomes, received FDA approval in 1995 and 1999, respectively, followed by many of the category [34,35]. Despite availability of around sixteen liposomal drugs in the market as of now, very few formulations are designated as the treatment modality for NSCLC. Here we summarize a few recent examples of liposomes for pulmonary delivery of therapeutics.

In 2014, Cheng et al. [36] exploited EGFR binding affinity of a novel peptide GE11 in doxorubicin-loaded liposomes and characterized them in terms of size dis­tribution, zeta potential, drug entrapment efficiency, and morphology. It turned out that optimal GE11 density was 10% in A549 cytotoxicity. Major involvement of clathrin-mediated endocytosis pathway was also determined by cellular uptake experiments. Using a near infrared (NIR) fluorescence imaging system, they found that the accumulation and retention of the GE-11 modified liposomes was 2.2-fold higher compared to unmodified liposomes [36]. For effective delivery of triptolide (TPL) to NSCLC by pulmonary administration, a dual ligand (anti-carbonic anhydrase IX (anti-CA IX) antibody and CPP33) modified triptolide-loaded liposomes (dl-TPL-lip) was designed, synthesized, and characterized by Lin et al. [37] in 2018. The cell killing ability was evaluated by an apoptosis assay. Importantly, superior tumor penetration and tumor growth inhibition efficacy of the liposomes were further demonstrated using 3D tumor spheroids. Pharmacokinetics studies in rats after endotracheal administration of the liposomal formulations exhibited a lower concentration of TPL in circulation [37]. In 2017, Song et al. [38] came up with a multifunctional targeting liposome for the treatment of NSCLC and achieved better in vivo effects. They decorated the liposome surface by Octreotide (OCT), a synthetic 8-peptide analog of somatostatin that binds to somatostatin receptors overexpressed in a variety of tumors. Two drugs were co-encapsulated in the liposome: Honokiol into the lipid bilayer to reduce tumor metastasis and inhibit vasculogenic mimicry channel formations, and epirubicin into the aqueous core as an antitumor drug. Mechanistic investigation studies revealed that these liposomes could downregulate PI3K, MMP-2, MMP-9, VE-Cadherin, and FAK and activate caspase 3 [38]. In 2013, Mukherjee et al. designed and synthesized a series of guanidinylated cationic amphiphiles and demonstrated that systemic administration of a 19 bp synthetic CDC20 siRNA encapsulated within liposomes of guanidinylated cationic amphiphile with stearyl tails inhibits B16F10 solid tumor growth and intravenous administration of the same liposomal formulation inhibits B16F10 melanoma growth on lung (metastases) in a syngeneic C57BL/6J mouse tumor model [39].

As a suitable alternative, pulmonary drug delivery strategy, i.e., local delivery via inhalation has gained substantial attention of researchers across the globe. This enables us to reach higher local drug concentration to the specific site of action. This also demands a low dose of therapeutics with reduced toxicity. We intend to provide few examples of liposomes to be inhaled as reported in the last two decades: In 2000, Anderson PM and coworkers designed a phase I study to examine the efficacy and toxicity of administering interleukin (IL)-2 liposomes by aerosol to patients with lung metastases. The liposome-aerosol was inhaled for ~20 min thrice a day. The dose chosen was based on previous studies. Nine patients were treated in three cohorts of three patients at varying IU of IL-2 thrice a day. Authors reported that inhalation of IL-2 liposomes is well tolerated with any reduced systemic toxicity [40]. In another phase I dose-escalating study performed by Wittgen et al. [41] safety, efficacy and PK of aerosolized liposomal (sustained release lipid inhalation targeting, SLIT) cis-platin were investigated in patients having lung cancer. In seventeen patients SLIT cis-platin was well tolerated. No nephrotoxicity, ototoxicity, hematologic toxicity, or neurotoxicity was revealed by safety data. Together, they concluded that the use of aerosolized liposomal cis-platin was feasible and safe [41]. Importantly, in 2010, Makale and co-workers developed a PEGylated DOX liposome with a dextran core, containing iron oxide for MRI contrast and Bodipy for fluorescence and demonstrated robust imaging ability of these nanoparticles (NPs) in an in vivo murine Lewis lung carcinoma model [42]. In 2011, Lowery et al. [43] decorated the surface of a DOX entrapped liposome with a phage displayed peptide HVGGSSV to achieve radiation guided selective drug delivery to tumors. Liposomes were labeled with Alexa Fluor 750 and the biodistribution of the labeled liposomes was studied in a murine Lewis lung carcinoma model by near infrared (NIR) imaging [43]. The liposomes are widely applied for lung cancer theranostics as they show immense promise due to their excellent biocompatibility and biodegradability [44,45]. Moreover, liposomes hold the edges over other nanoparticles as they are useful to load a high amount of therapeutic agents and can be easily controlled for sustained drug delivery [44,45]. However, liposomal nanoparticles have some disadvantages like batch to batch variability, high manufacturing cost, possible drug leakage, etc. [44]. Hence, careful fabrication strategies are required to overcome these challenges and reduction of the cost.

## 5. Polymeric Nanoparticles for Lung Cancer Theranostics

Polymeric NPs, on the other hand, can be prepared either by nanoprecipitation or a double emulsion method via self-assembly of biodegradable amphiphilic block-copolymers with varying hydrophobicity’s between blocks and are suitable for systemic administration. The core-shell structure of polymeric NPs facilitates encapsulation of hydrophobic drugs, extension of circulation time, and sustained drug release. Their surfaces can also be decorated for targeted drug delivery [46,47]. For instance, Genexol-PM is a formulation of paclitaxel and poly (D,L-lactide)-b-polyethylene glycol-methoxy (PLGA-mPEG), which is already marketed for metastatic breast cancer therapy in Korea and other European countries [48,49]. Here we summarize a few recent interesting examples of usage of polymeric NPs for treatment of lung cancer.

In 2015, Jiang [50] developed a nano-carrier encapsulating Crizotinib (approved for EML4-ALK fusion positive lung cancer) within polylactide tocopheryl polyethylene glycol 1000 succinate (PLA-TPGS), which showed a sustained release, induced remarkable cytotoxicity in NCIH3122 lung cancer cells, and noticeable early and late apoptosis. The polymeric nanoparticle followed an endocytosis-mediated cellular uptake [50]. Interestingly, in 2017 Hu et al. [51] reported on the efficiency of paclitaxel (PTX) loaded Polycaprolactone/ Poly (ethylene glycol)/Polycaprolactone (PCEC) nanoparticles combined with chronomodulated chemotherapy for use in lung cancer. The authors set out to map the suitable time of the day for administering drug loaded nano-carriers by making out the crucial role of circadian rhythms in cancer propagation. The combination therapy demonstrated remarkable tumor growth inhibition in vivo, while it turned out that 15HALO is optimal for chemotherapy [51]. Very recently, to circumvent the low targeting capacity of nanoparticles, Wang et al. used mesenchymal stem cells (MSC) as a carrier for drug delivery loaded with nanoparticles with docetaxel (DTX). MSC proved its superiority over fibroblasts in drug loading. Both cellular and animal experiments justified the intercellular translocation of nanoparticles from MSC to cancer cell. It also inhibited primary tumor growth in vivo [52]. Much to our intrigue, Ganesh et al. explored HA-PEI/PEG nano-carriers for CD44-targeted siRNA delivery to lung cancer cells. They undertook a detailed structure-activity study for optimal siRNA encapsulation efficiency. Importantly, the targeted HA-PEI/PEG nanosystems encapsulating SSB/PLK1 siRNA showed higher cellular uptake and sequence specific gene knockdown in vivo both in sensitive and resistant A549 primary and metastatic [53].

Polymer based delivery of chemotherapeutics via inhalation has also made substantial progress in recent years. Kim et al., in 2013, made an attempt to make highly porous PLGA microparticles entrapping doxorubicin where the surface of the particles was decorated with Apo2L/TRAIL (tumor necrosis factor (TNF)-related apoptosis-inducing ligand) that binds selectively to death receptors such as DR4/TRAIL-R1 expressed specifically on cancer cells. The particles got deposited in a mouse lung followed by pulmonary administration and exhibited tumor growth inhibition in nude mice with H226 metastatic lung cancer cells [54]. Furthermore, for pulmonary inhalation treatment, Feng et al. developed a two-in-one nanosystem comprising of doxorubicin and paclitaxel encapsulated into a porous PLGA microparticles and established a synergy between the two drugs at DOX:PTX 5:1 in vitro. Co-delivery remained superior over their individual counterparts in vivo as well [55]. Khatun et al. [56] demonstrated multifunctional cancer theranostics application of graphene-doxorubicin in a HA nanogel in human lung cancer cell line (A549). This nanocomposite was used as thermo and chemotherapeutic, real-time noninvasive optical imaging, and a controlled drug release [56] (Figure 2). Importantly, in a study performed in 2006, Mitra et al. [57] observed an enhanced tumor targeting the capability of N-(2-hydroxypropyl) methacrylamide (HPMA) copolymers when coupled with RGDfK and RGD4C in C57BL/6NHsd mice bearing Lewis lung carcinoma. They conjugated radionuclides with longer half-life such as 111In compounds to HPMA for scintigraphic imaging and detected contrast enhancement at the tumor site after 24h of injection [57]. In 2010, Gao and co-worker developed an MRI-visible PLA polymeric micelle whose surface was decorated by a peptide RGDLATLRQL directed towards αvβ6 integrin over-expressing human NSCLC cell H2009. DOX and super-paramagnetic iron oxide nanoparticles (SPIONs) were loaded inside the micelle core for drug delivery and MR imaging, respectively [58]. The polymeric nanoparticles are extensively applied for lung cancer theranostics due to their biocompatibility, biodegradability, high drug pay load, sustained drug release, high scalability, low batch variability, low cost production, and easy tunability [44,45].

## 6. Bio-Nanoparticles for Lung Cancer Theranostics

Due to high biocompatibility, better stability and biodegradability current researchers have shifted their focal point of research towards using the bio-nanoparticles including protein nanoparticles, solid lipid nanoparticles, viral nanoparticles, aptamers, and apoferritin, wherein a bio-mimicking component is incorporated to the therapeutic nanoparticles [59,60]. In the recent past, these types of nanoparticles were successfully designed, synthesized, and utilized for cancer theranostics applications in lung cancer [59,61,62].

### 6.1. Viral Nanoparticles

Viral nanoparticles (VNPs) obtained from viruses and bacteriophages have gained immense interest for various biomedical applications including drug delivery, biosensing, bioimaging, and vaccine development due to their biocompatibility, flexibility in sizes and shapes, and easy surface modification [63]. Many researchers have developed a combinational approach with chemotherapy and immunotherapy for the treatment of lung cancer due to challenges of drug resistance. Veljanski et al. has published an exciting review about the use genetically modified oncolytic viruses (OVs) with conventional chemotherapies in the treatment of lung cancer. The inability of chemo-drugs to destroy the cancer stem cells is well compensated by OVs-based gene therapy [64]. The cowpea mosaic virus (CPMV) with an average size of ~28 nm has a high potential for vaccination therapy in lung cancer [65].

### 6.2. Protein-Based Nanoparticles

Protein nanoparticles prepared from a naturally occurring protein, such as gelatin, gliadin, albumin, and legumin have been recently used for the drug and gene delivery purposes either alone or in a mixture with biodegradable polymers in lung cancer therapy due to their excellent biocompatibility, and lack of inflammation in human bronchial cells and high cellular uptake [66,67,68]. Cationic bovine serum albumin (CBSA) has been utilized for the delivery of siRNA for the metastatic lung cancer therapy [69].

### 6.3. Apoferritin

Apoferritin is the hollow protein nanocage without the iron core that is composed of self-assembling 24 polypeptide subunits and has internal and external diameters of 8 nm and 12 nm, respectively. Upon removal of the iron core the apoferritin undergoes a process of assembly and disassembly with the change in pH that is extensively utilized for the synthesis of various nanoparticles for lung cancer theranostics [70,71]. Li et al. demonstrated the lung cancer diagnosis of A549 cells by using fluorescence and MR imaging of apoferritin, a ferritin-based multifunctional nanostructure [71]. The authors synthesized multifunctional hybrid nanostructures made of ferritin that showed green fluorescence and had ferrimagnetic iron oxide nanoparticles into the hollow ferritin cavity. This multifunctional apoferritin was used for the imaging αvβ3 integrin upregulated cancer cells. In another recent paper by Luo et al., demonstrated the use of hyaluronic acid (HA)-conjugated apoferritin nanocages for pH-responsive controlled intracellular prodrug release of the anticancer drug daunomycin (DN), which was encapsulated into the interior of apoferritin [72]. Moreover, the authors modified the apoferritin by HA to target and kill the cancer cells (embryonic lung MRC-5 cells and lung cancer A549 cells) upon binding to the HA-receptor CD44.

Bio-nanoparticles are widely applied for lung cancer theranostics as they demonstrate huge promise due to their good biocompatibility and biodegradability. However, the synthesis strategies can be complex at times and that can increase the cost and time of manufacturing. Hence, more research needed to manufacture these bio-nanoparticles from lab-scale to commercial industrial scale.

## 7. Inorganic Nanoparticles for Lung Cancer Theranostics

Inorganic nanoparticles have long been used for various biomedical applications including drug delivery, nucleic acid delivery, bio-sensing, diagnostics, imaging, and cancer therapy due to their exciting physico-chemical properties in the nanoscale range [73,74,75,76]. Among various inorganic nanoparticles, gold, silver, iron oxides, silica, rare earth oxides, carbon dots, and nanodiamonds were extensively studied in lung cancer theranostics [77,78,79]. Moreover, these nanoparticles demonstrated prominent cytotoxic effects on various lung cells in vitro and in vivo that depend largely on their size, shape, surface charge, concentration, and time of exposure. An accurate control over these physico-chemical parameters can facilitate their meaningful application in lung cancer theranostics.

### 7.1. Gold Nanoparticles (AuNPs)

Gold nanoparticles were extensively used for cancer theranostics applications due to easy synthesize and functionalize high biocompatibility, and multifunctional theranostics properties [80,81,82]. Several research groups utilized AuNPs for lung cancer theranostics [77]. For example, Nanospectra has developed a silica-gold nanoshells stabilized by (poly)ethylene glycol (PEG) for the photothermal therapy to the solid tumors using an NIR light source [83]. More importantly, in a recent clinical trial, AuroLase^®^ was used for the photothermal therapy of primary or metastatic lung tumors (NCT01679470) [84]. In a recent published report by Knights et al. the authors demonstrated the size dependent effects of gold nanorods (AuNRs) on the photoacoustic (PA) imaging response and pulsed-wave photothermal therapeutic (PW-PPTT) efficacy, which is crucial for the clinical translation of AuNRs [77]. Interestingly, the PA intensity increased with the AuNR size due to the overall mass of the nanoparticles. All the different sized AuNRs showed toxicity in lung cancer cells upon laser fluence with a highest cell death in the smallest AuNR treatment, indicating the theranostics potential of AuNRs combined with PW lasers in lung cancer. In another recent report, Ramalingam et al. [85] showed the enhanced anti-cancer efficacy of doxorubicin (DOX) using polyvinylpyrrolidone functionalized AuNPs (Dox@PVP-AuNPs) in lung cancer cells. Mechanistic studies demonstrated that Dox@PVP-AuNPs treatment to lung cancer cells increases reactive oxygen species (ROS) generation, up-regulates the tumor suppressor genes, sensitize mitochondrial membrane potential and further induces apoptosis [85]. Peng et al. exhibited that novel sensors based technology using gold nanoparticles could be used for a non-invasive and inexpensive diagnostic tool for lung cancer [86].

### 7.2. Iron Oxides Nanoparticles (IONPs)

Supermagnetic iron oxide nanoparticles is extensively used as a MRI contrast agent, and also can be utilized as a delivery carrier in cancer theranostics applications. Iron oxides nanoparticles were long used for various biomedical applications including MRI imaging, drug delivery, magnetic hyperthermia, and cancer theranostics in lung cancer [87,88,89,90]. Wang et al., recently demonstrated MRI and magnetic resonance-guided focused ultrasound ablation therapy using iron oxide nanoparticles in lung cancer (Figure 3) [87]. The authors synthesized an epidermal growth factor receptor targeted PEGylated iron oxide nanoparticles (IONPs) for targeted delivery and imaging to lung cancer in an in vitro and in vivo rat xenograft model of human lung cancer (H460). In another study, Sadhuka et al. showed the usage of EGFR-targeted inhalable iron oxide nanoparticles for magnetic hyperthermia in lung cancer [89]. The authors showed that EGFR targeting enhanced the tumor retention of IONPs. Moreover, magnetic hyperthermia treatment by EGFR-targeted IONPs caused in major inhibition of lung tumor growth in in vivo orthotropic lung cancer model.

### 7.3. Silver Nanoparticles (AgNPs)

Silver nanoparticles were long used for various biomedical applications including anti-bacterial applications, anti-cancer applications, fluorescence imaging, and biosensors [91]. Recently, He et al. demonstrated the antitumor activity of biosynthesized AgNPs against lung cancer in in vitro H1299 lung cancer cells and an in vivo xenograft immunodeficient (SCID) mouse model [92]. The potent cytotoxicity effect of AgNPs was showed by trypan blue and MTT assay. Mechanistic studies showed that AgNPs caused apoptosis (increase in caspase-3 and decrease in bcl-2) in lung cancer cells that connected well with an inhibition of NF-κB activity [92]. The cytotoxicity of AgNPs largely depends on their size, shape, morphology, and surface chemistry [93]. In another recent published report, Jeong et al. [94] explored the basic mechanism of hypoxia on AgNPs induced apoptosis that showed the upregulation of HIF-1α expression under both normoxic and hypoxic conditions. Moreover, the AgNPs treatment caused programmed cell death in lung cancer cells but not in normal cells. Notably, HIF-1α protected AgNPs-induced apoptosis by regulating autophagic flux through controlling the ATG5, p62, and LC3-II. Hence, these results suggest that hypoxia-mediated autophagy could be used to inhibit the AgNPs mediated apoptosis involving the HIF-1α as a potential target in lung cancer therapy [94].

### 7.4. Other Metal Nanoparticles

Among other nanoparticles rare earth, silica, and nanodiamond were used for the cancer theranostics application in lung cancer [95,96,97]. Wu et al., showed the application of a silica-polymer nanocomposite for p53 gene therapy and near infrared tumor targeted imaging of lung cancer in vitro and in vivo [95]. A nanodiamond (ND) was also utilized for the delivery of paclitaxel in lung cancer therapy. This nanodrug delivery exhibited significant tumor regression ability in immunodeficiency mice, in a lung cancer cell model. Mechanistic studies revealed that ND caused mitotic arrest and apoptosis that led to lung cancer cell death [97]. In another report, Zhang et al. developed a silica based imaging agent to detect a single miRNA in lung cancer cells that can be used for biosensing applications [96]. In another published report, Chen et al. showed the application of micromolar concentration of the neodymium oxide nanoparticles (Nd_2_O_3_) for the induction of extensive autophagy and immense vacuolization in NSCLC cells [98]. Wu et al. [99] developed a multi-functionalized, carbon dots based theranostics nanoagent that can be used for bioimaging as it emitted visible blue photoluminescence when excited at 360 nm and also can be utilized as a gene delivery vehicle for multiple siRNAs (EGFR and cyclin B1) in lung cancer. Moreover, this nanoagent was found to be accumulated in lung cancer cells by receptor mediated endocytosis in a targeted manner, resulting in improved gene silencing and anti-cancer efficacy.

Scientists and researchers are excited with metal nanoparticles in the recent past due to their small size, high surface to area, easy synthesis and scale up, low cost, multifunctional theranostics applications, exciting physico-chemical properties. However, due to the lack of enough knowledge about their long-term toxicity, pharmacokinetics, pharmacodynamics, and degradability, the clinical translation of these nanomaterials is still not widely possible compared to its other counterparts [29].

## 8. Clinical Status of the Nanotheranostics in Lung Cancer

Advances in nanoparticle designing and formulations are related to numerous applications in the detection and treatment of malignant growth. Ongoing advances in nanotheranostics in imaging and therapeutics of lung cancer are as follows. Porphyrins due to their favorable photophysical properties have been particularly successful for cancer imaging and photodynamic therapy (PDT). Hematoporphyrin derivative (HpD), porfimer sodium has been approved in over 120 countries since 1993 for the detection and photodynamic therapy of esophageal, lung, superficial bladder, gastric, cervical, and endobronchial cancers [100,101,102]. Though relatively non-toxic, HpD is not very effective as a primary therapy in lung cancer [103]. Laserphyrin 664 is approved for PDT in Japan for treating early centrally located lung cancer. PET (positron emission tomography) is a nuclear medicine imaging method, frequently used in combination with FDG (fluorine-18 combined with deoxy-glucose) [104]. FDG is a glucose analogue that is extensively used in cancer staging, restaging and for the analysis of tumor response to treatment. The most frequent application of theranostics is for the palliative treatment of bone metastases from lung cancer. EDTMP marked with 153Sm [105] and 99mTc-MDP an analogue of pyrophosphate similar to bone scanning agents and provides high doses of localized radiation because of its β-particle emissions.

Peptidomimetics like [68Ga]Ga-NODAGA-THERANOST™ is a 3 integrin antagonist first used in humans for lung and breast cancer diagnosis and anti-angiogenic therapy [106]. Ongoing advances in nanotheranostics for drug delivery have been mostly attempted using liposomal formulations. Irinotecan liposome injection (ONIVYDE®) is being investigated versus topotecan in patients with small cell lung cancer after platinum-based first-line therapy (Phase3; 2018–2022 study). To find the highest dose of DOTAP:Chol-fus1 liposomal formulation that can safely be given in combination with tarceva (erlotinib hydrochloride) to patients with NSCLC (Phase2; 2014–2019 study) is currently undergoing. Another trial is investigating the pain management efficiency of liposomal bupivacaine after elective thoracoscopic lobectomy (Phase3; 2018–2021 study) for non-small cell lung cancer.

An exosome is a nano-sized vesicle secreted from different cell types and has a natural ability to carry functional biomolecules, such as small RNAs, DNAs, and proteins in their lumen. This unique signature of particular cells also makes them attractive for use in drug delivery and molecular diagnosis. Moreover, exosomes can be coupled to nanoparticles and used for high precision imaging. Exosomes are considered as an important component in liquid biopsy assessments, which are useful for detecting cancers, including lung cancer. Several studies are currently underway to develop methods of exploiting exosomes for its use as efficient drug delivery vehicles and to develop novel diagnostic modalities. Srivastava et al., 2018 combined the diagnosis of CT and exosome in early lung cancer 2018–2019. Another theranostics-based use of exosome is for the dynamic monitoring circulating tumor DNA in surgical patients with lung cancer, 2017–2023 [107]. Albumin based nanoparticles are in use in different phases of clinical trials for the delivery of paclitaxel, cisplatin in NSCLC [62,108,109,110].

The development of lung cancer nanotheranostics, with an accentuation on clinical use is limited by several lacunae. The reproducible synthesis of several nanoplatforms with composite structures and its scale-up produces significant complications. Sterility can also hamper such processes. Another important aspect worth considering is nanoparticles-induced cytotoxicity, genotoxicity, and immunotoxicity related to their nanometer size. Industrial vendors and pharmaceutical companies have fewer incentives to move on with theranostics-based blockbusters because of the projected final product price as compared to conventional therapy. Though, coupling therapy with diagnosis together into a single theranostics platform should provide significant advantages but the imaging and tumor targeting components might have different pharmacokinetics and dynamics and can mount a significant manufacturing challenge.

Delay in the transition for theranostics from bench to bedside is mainly due a lack of basic-clinical scientific collaborations. Basic scientists and many clinical scientists lack the information or time to comprehend, envision, oversee cons, and manage the nuances of the multistep compliance guidelines of the clinical trial phases. Being oblivious about the obligation regarding guaranteeing consistency with the guidelines and related regulatory aspects delays the process of bringing cancer theranostics from the laboratory to clinical use. Though FDA is bringing a change in the regulatory changes to the regulatory procedures and developing tools to reduce the development time of theranostics. Weighing the potential benefits to possible adverse health risk assessment of lung cancer nanotheranostics needs to be considered based on scientifically sound, evidence-based and well-controlled clinical studies.

## 9. Conclusions and Future Perspectives

Theranostics or personalized cancer treatment is proof based, an individualized prescription that warrants the right treatment at the correct time, leading to significant efficacy and improvements of patient’s condition and a decrease in medicinal and services costs. Theranostics medicine is employed for delivering both therapeutic and imaging agents to the targeted area of the body, using a nanotechnology-based delivery platform. Nanomaterials have proved to be tools with tremendous benefit and are now finding application in the clinic. Nanostructures, due to their novel physical properties can often overcome solubility and stability issues through surface modification/wrappings or additional formulation. The integrated approach of combining ligands, drugs, biomolecules, and imaging agents into a functionalized nanoparticle enables targeted drug delivery and diagnostics. Another aspect that aids in the higher therapeutic payload is the high surface area because of their nanosize. Nanoparticle-based targeting specific cancer cells and specific delivery of therapeutic payloads at cancer sites via passive or active targeting can significantly reduce nonspecific toxicity. In spite of the advantages, a lot of unsolved challenges remain including scale-up problems, economical production, the pharmacokinetics of the drug, and the imaging construct. Additional issues with nanotoxicity and regulatory guidelines and hurdles need to be resolved in order to see lung cancer theranostics in the clinic. Though nanotechnology has achieved great strides but still is not used to maximal impact in lung and other malignancies.

With the advent of stronger sequencing, immunohistochemistry, and proteomic techniques, a better understanding of the mechanisms of cancer and identification of new definitive biomarkers are on the way. Greater funding for multi-center cohort studies and the advent of landmark genomic program like the Cancer Genome Atlas (TCGA) and cancer proteome studies the human protein atlas have also created new inroads in the understanding of cancer. In 2018, theranosticsimaging using PSMA (prostate specific membrane antigen) PET (positron emission tomography) and image-guided therapy using PSMA targeted radionucleotide lutetium-177 PSMA617 has shown to be a paradigm-changing practice for improving prostate cancer patient outcomes. The advances of theranostics-based radiomics will span from the current effort to the expected future of using deep learning. Development of theranosticstools for ultrasensitive and quantitative measurement of theranostics biomarkers, ability to diagnose and quantify cancer at its earliest stage with high resolution, and early prediction response to cancer therapy will also depend on how new assisting tools (e.g., artificial intelligence) are going to be harnessed. The integration of all the advances in these allied fields together as cancer theranosticsis poised to revolutionize the future of therapeutics for the ultimate eradication of cancer.

## Figures and Tables

**Figure 1 cancers-11-00597-f001:**
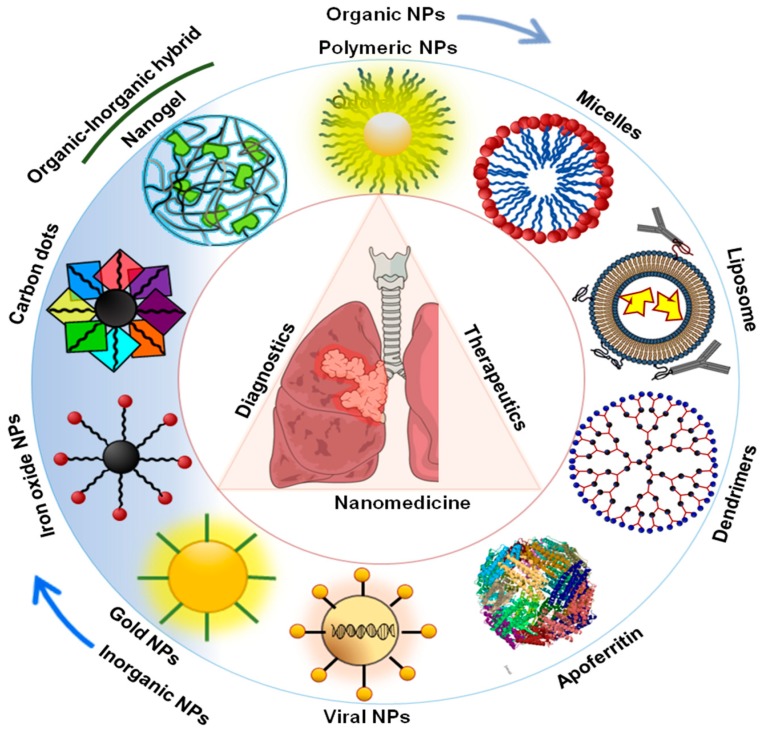
Schematic representation of different theranostics nanomedicine approaches in lung cancer theranostics.

**Figure 2 cancers-11-00597-f002:**
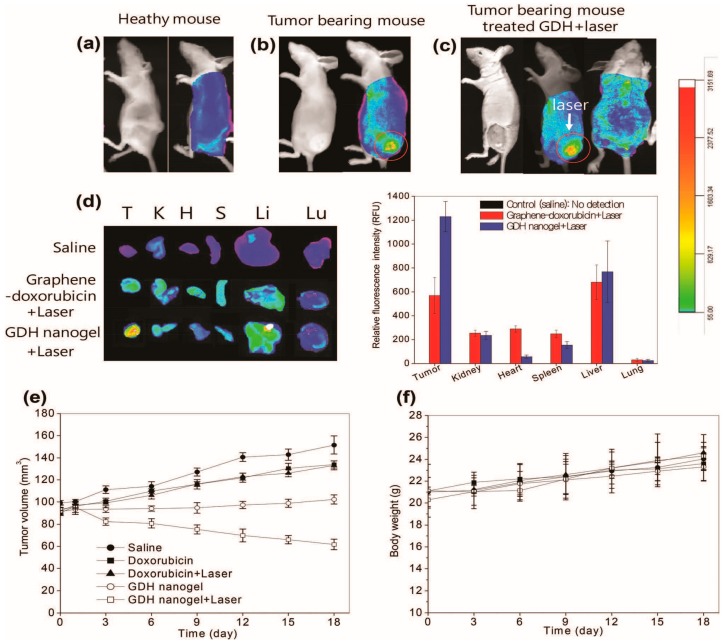
In vivo optical imaging and thermo-chemotherapy using the nanogel. (**a**,**b**) are optical images of healthy mice and tumor bearing mice, respectively. (**c**) Light responsive imaging. The nanogels were intravenously injected and 670 nm laser was applied to the tumors for 30 min. (**d**) Ex vivo imaging and fluorescence intensities of tumors and normal tissues. Organs were arranged in the following order: Tumor (T), kidney (K), heart (H), spleen (S), liver (Li), and lung (Lu). (**e**) Thermo-chemotherapy after treating doxorubicin and the nanogels with and without laser irradiation. (**f**) Body weight changes of mice after treatment with the nanogels. The data were plotted as mean ± SEM (*n* = 5). Reproduced from [56]. Copyright © 2015 Royal Society of Chemistry.

**Figure 3 cancers-11-00597-f003:**
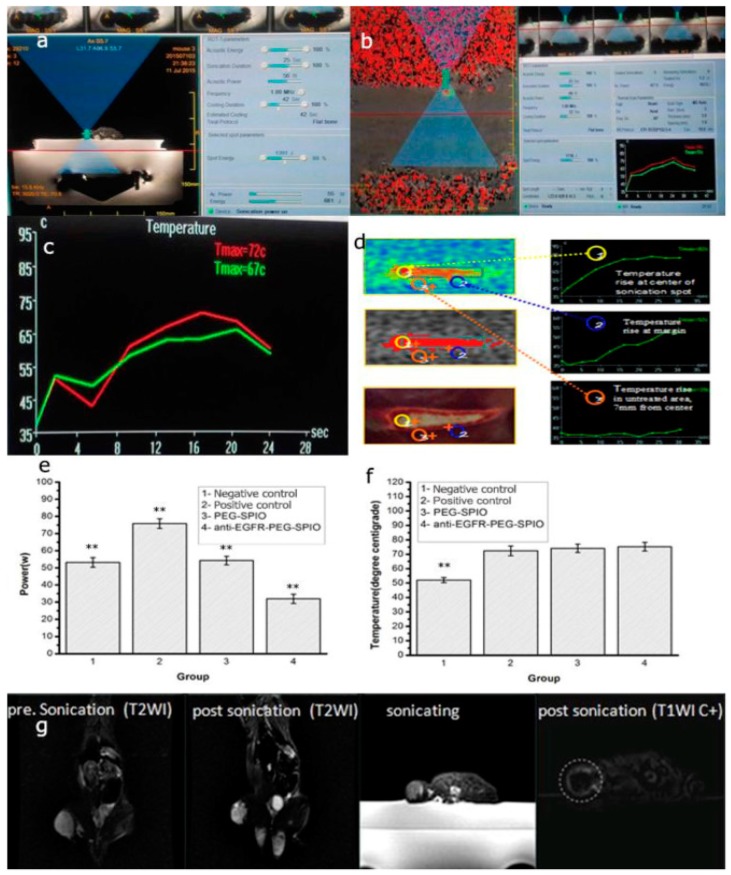
(**a**) Sonication process images. Treatment planning software on the MRgFUS workstation. In total, five sonications are needed to measure the circumference of this tumor. (**b**) Tissue temperature mapping during MRgFUS ablation. (**c**) Real-time temperature change in MRgFUS ablation monitored by MRI. (**d**) Schematic illustration of therapeutic temperature map in tumor center and margin. Sonication energy (**e**) and therapeutic peak temperature (**f**) of negative control (low power, 54 W), control (76 W), PEG-SPIO (54 W), and anti-EGFR-PEG-SPIO (32 W) at 4 h post-injection of SPIO nanoparticles. (**g**) Anti-EGFR-PEG-SPIO group: Coronal T2WI image signal intensity of tumor before treatment and after treatment. The coronal T2WI signal intensities of tumor increased significantly after therapy compared to before therapy (Figure 6g). Enhanced axial T1WI-weighted images after injection of Gd-DTPA. Axial contrast-enhance T1WI subtraction images after injection of Gd-DTPA showed a small focal area of nonperfusion. Reproduced from [87]. Copyright © 2017. Elsevier.

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
