# Peer review of "Recent Progress in the Theranostics Application of Nanomedicine in Lung Cancer"

_cancers, 2019, doi:10.3390/cancers11050597_

Round 1

Reviewer 1 Report

The Article is well written and does summaries the current progression in Lung cancer theranostics using nanomedicine. I would strongly recommend for its acceptance after a minor english correction and spell checks.

Author Response

We thank the reviewer for giving constructive comments. We have corrected the minor English and grammatical corrections all throughout the revised manuscript.

Reviewer 2 Report

The authors here summarize various nano carrier based theranostics for lung cancer.

This is an interesting read. 

The introduction how ever claims to summarize the nano therapeutics for lung cancer all together, rather than just theranostics. The authors might have to modify that.

The review needs to be re organized to continue a good flow to the read.

Adding a some more introduction about how theranostics are an important advancement will be great.

It will also be informative, if the authors can provide the disadvantages of the systems immediately when they are first cited and talk about the current gaps in literature and what can be done to improve them.

Author Response

Comment 1: 

The authors here summarize various nano carrier based theranostics for lung cancer.

This is an interesting read. 

The introduction how ever claims to summarize the nano therapeutics for lung cancer all together, rather than just theranostics. The authors might have to modify that.

Reply: We thank the reviewer for his/her valuable comments. We have now modified the introduction to discuss more about nanotheranostics and lung cancer.

Comment 2: The review needs to be re organized to continue a good flow to the read.

Reply: We thank the reviewer for proving constructive comments. We have re-organized the points and now think that it gives a better flow for reading.

Comment 3: Adding a some more introduction about how theranostics are an important advancement will be great.

Reply: We are grateful for the suggestion. We have now added an extra paragraph about the theranostics achievement in the introduction part of the revised manuscript. 

Comment 4: It will also be informative, if the authors can provide the disadvantages of the systems immediately when they are first cited and talk about the current gaps in literature and what can be done to improve them.

Reply: We thank reviewer for his/her constructive comments. We have now included the disadvantages of each systems right after the sections for better reading.